# Activation of Serum/Glucocorticoid Regulated Kinase 1/Nuclear Factor-κB Pathway Are Correlated with Low Sensitivity to Bortezomib and Ixazomib in Resistant Multiple Myeloma Cells

**DOI:** 10.3390/biomedicines9010033

**Published:** 2021-01-04

**Authors:** Masanobu Tsubaki, Tomoya Takeda, Takuya Matsuda, Shiori Seki, Yoshika Tomonari, Shoutaro Koizumi, Miki Nagatakiya, Mai Katsuyama, Yuuta Yamamoto, Katsumasa Tsurushima, Toshihiko Ishizaka, Shozo Nishida

**Affiliations:** 1Division of Pharmacotherapy, Kindai University School of Pharmacy, Kowakae, Higashi-Osaka 577-8502, Japan; tsubaki@phar.kindai.ac.jp (M.T.); takeda@phar.kindai.ac.jp (T.T.); takuya.matsuda.kindai@gmail.com (T.M.); seki_kindai@yahoo.co.jp (S.S.); tomonari_kinki@yahoo.co.jp (Y.T.); koizumi.s.kindai@gmail.com (S.K.); nagatakiya.m.kindai@gmail.com (M.N.); sgvja70235@yahoo.co.jp (M.K.); yamamoto_kindai@outlook.jp (Y.Y.); tsurushima_kindai@yahoo.co.jp (K.T.); 2Department of Pharmacy, Sakai City Medical Center, Sakai 593-8304, Japan; ishizaka_sakai@yahoo.co.jp

**Keywords:** multiple myeloma, bortezomib, ixazomib, SGK1, nuclear factor (NF)-κB, low sensitivity

## Abstract

Multiple myeloma (MM) is an incurable malignancy often associated with primary and acquired resistance to therapeutic agents, such as proteasome inhibitors. However, the mechanisms underlying the proteasome inhibitor resistance are poorly understood. Here, we elucidate the mechanism of primary resistance to bortezomib and ixazomib in the MM cell lines, KMS-20, KMS-26, and KMS-28BM. We find that low bortezomib and ixazomib concentrations induce cell death in KMS-26 and KMS-28BM cells. However, high bortezomib and ixazomib concentrations induce cell death only in KMS-20 cells. During Gene Expression Omnibus analysis, KMS-20 cells exhibit high levels of expression of various genes, including *anti-phospho-fibroblast growth factor receptor 1 (FGFR1)*, *chemokine receptor type (CCR2)*, and *serum and glucocorticoid regulated kinase (SGK)1*. The SGK1 inhibitor enhances the cytotoxic effects of bortezomib and ixazomib; however, FGFR1 and CCR2 inhibitors do not show such effect in KMS-20 cells. Moreover, SGK1 activation induces the phosphorylation of NF-κB p65, and an NF-κB inhibitor enhances the sensitivity of KMS-20 cells to bortezomib and ixazomib. Additionally, high levels of expression of SGK1 and NF-κB p65 is associated with a low sensitivity to bortezomib and a poor prognosis in MM patients. These results indicate that the activation of the SGK1/NF-κB pathway correlates with a low sensitivity to bortezomib and ixazomib, and a combination of bortezomib and ixazomib with an SGK1 or NF-κB inhibitor may be involved in the treatment of MM via activation of the SGK1/NF-κB pathway.

## 1. Introduction

Multiple myeloma (MM), the second most common hematological cancer, is defined as the monoclonal proliferation of plasma cells in bone marrow and the discharge of monoclonal immunoglobulins [1]. The incidence of MM increased worldwide from 1990 to 2016. The 5-year survival rate for patients with MM is now 50%, owing to the development of novel therapeutic agents, such as immunomodulatory drugs, histone deacetylase inhibitors, proteasome inhibitors, and monoclonal antibodies [1,2]. However, in a fairly large proportion of cases, MM is an incurable malignancy because it often presents with primary and acquired resistance to therapeutic agents [3]. Thus, it is important to elucidate the primary and acquired resistance mechanisms of MM cells.

Bortezomib, the first proteasome inhibitor, binds to the caspase- and chymotrypsin-like active sites of the 20S proteasome, suppresses proteasome activity, and induces apoptosis associated with the endoplasmic reticulum (ER) stress response [4,5]. Ixazomib, the first orally utilizable proteasome inhibitor, is administered in combination with dexamethasone and lenalidomide in patients with refractory and relapsed MM, and this combination regimen increases progression-free survival in patients who have standard- and high-risk MM [6,7]. The development of bortezomib resistance is associated with a poor prognosis, and the median overall survival of patients with bortezomib-resistant MM is nine months [8]. Mutations in the β5 proteasome subunit, a binding pocket for proteasome inhibitors, are correlated with bortezomib and ixazomib resistance in MM cells but these mutations are nearly undetectable in MM with primary and acquired resistance to bortezomib [9,10]. Additionally, the overexpression of the β5, β2, and β1 subunits induces bortezomib resistance in MM cells [11,12,13]. Patients with refractory and bortezomib-resistant MM exhibit the downregulated expression of X-box binding protein 1 (XBP1), an ER stress regulator, and loss-of-function mutations of XBP1 promote bortezomib resistance in patients with MM [14,15]. Furthermore, the constitutive activation of nuclear factor-κB (NF-κB) and bone marrow stromal cell-induced NF-κB activation are involved in bortezomib resistance in MM cells and primary MM cells [16,17].

Serum and glucocorticoid regulated kinase 1 (SGK1) are family members of cyclic adenosine monophosphate (cAMP)-dependent, cyclic guanosine monophosphate (cGMP)-dependent, and protein kinase C (AGC) kinases, and regulates ion channels, transporters, transcription factors, and enzymes [18]. SGK1 is overexpressed in myeloma, medulloblastoma, prostate, colon, ovarian, and non-small cell lung cancer, and induces cell proliferation, survival, and drug resistance [18]. Regarding MM, activation of SGK1 by interleukin 6 (IL-6) stimulation promotes cell cycle progression and DNA synthesis [19] and prevents endoplasmic reticulum-induced apoptosis by bortezomib [20]. However, the details of the mechanism of the SGK1-induced bortezomib- and ixazomib-resistance are not well understood.

Here, we investigate the mechanism underlying the primary resistance to bortezomib and ixazomib (low sensitivity to bortezomib and ixazomib) and aim to determine whether alternative small-molecule inhibitors can be used to enhance the cytotoxicity of bortezomib and ixazomib to MM cells.

## 2. Materials and Methods

### 2.1. Materials

Bortezomib, ixazomib, GSK650394, and PD166866 were purchased from SelleckChem (Houston, TX, USA). A C-C chemokine receptor type 2 (CCR2) antagonist was obtained from Santa Cruz Biotechnology (Dallas, TX, USA). Dimethyl fumarate (DMF) was purchased from FUJIFILM Wako (Tokyo, Japan). These reagents were dissolved in dimethyl sulfoxide (DMSO) and diluted in phosphate-buffered saline (0.05 M, pH 7.4).

### 2.2. Cell Culture

KMS-20, KMS-26, and KMS-28BM cells were obtained from the Japanese Collection of Research Bioresources Cell Bank (Osaka, Japan). These cells were maintained in RPMI 1640 medium (Sigma, St. Louis, MO, USA) supplemented with 25 mM of 4-(2-hydroxyethyl)-1-piperazineethanesulfonic acid (pH 7.4; FUJIFILM Wako), 100 U/mL streptomycin (Gibco, Carlsbad, CA, USA), 100 μg/mL penicillin (Gibco), and 10% fetal bovine serum (Gibco) in an atmosphere of 5% CO_2_.

### 2.3. Cell Viability

The effect of bortezomib, ixazomib, GSK650394, PD166866, C-C chemokine receptor type 2 (CCR2) antagonist, and dimethyl fumarate (DMF) on cell viability was assessed by the trypan blue stain exclusion assay as previously described [21,22,23,24].

### 2.4. Western Blotting

Cellular cytoplasmic and nuclear fractions were extracted using the ProteoExtract Subcellular Proteome Extraction Kit (Calbiochem, San Diego, CA, USA). These fractions were resolved by sodium dodecyl sulfate-polyacrylamide gel electrophoresis and transferred onto polyvinylidene fluoride membranes (GE Healthcare, Buckinghamshire, UK). The membranes were incubated with the following primary antibodies: anti-proteasome 20S subunit β5 (PSMB5) antibodies (Abcam, Tokyo, Japan), anti-phospho- serum and glucocorticoid regulated kinase (SGK)1 antibodies, anti-SGK1 antibodies, anti-phospho-fibroblast growth factor receptor 1 (FGFR1) antibodies, anti-FGFR1 antibodies, anti- chemokine receptor type (CCR)2 antibodies, anti-phospho-extracellular regulated protein kinase 1/2 (ERK1/2) (Thr202/Tyr204) antibodies, anti-ERK1/2 antibodies, anti-phospho-nuclear factor (NF)-κB antibodies (Ser536), anti-NF-κB antibodies, anti-phospho-Akt antibodies (Ser473), anti-Akt antibodies, anti-phospho-c-Jun N-terminal kinase (JNK) antibodies (Thr183/Tyr185), anti-JNK antibodies, anti-survivin antibodies, anti-X-linked inhibitor of apoptosis protein (XIAP) antibodies (Cell Signaling Technology, Beverly, MA, USA), anti-B-cell lymphoma-2 (Bcl-2)-associated X (Bax) antibodies, anti-Bcl-2-like protein 11 (Bim) antibodies, anti-Bcl-2 antibodies, anti-B-cell lymphoma-extra large (Bcl-xL) antibodies, anti-phorbol-12-myristate-13-acetate-induced protein 1 (Noxa) antibodies, anti-p53 upregulated modulator of apoptosis (Puma) antibodies, anti-lamin A/C antibodies (Santa Cruz Biotechnology), and anti-β-actin antibodies (Sigma). The membranes were then treated with horseradish peroxidase-coupled sheep anti-rabbit immunoglobulin G (GE Healthcare) and were visualized using Luminata Forte (Merck Millipore, Nottingham, UK).

### 2.5. Quantitative Real-Time Polymerase Chain Reaction (PCR)

Total RNA was extracted using RNAiso (Takara Biomedical, Shiga, Japan) and complementary deoxyribonucleic acid (cDNA) was synthesized from purified total RNA using the PrimeScript RT Master Mix (Takara Biomedical). cDNA was subjected to quantitative real-time Polymerase Chain Reaction (PCR) using SYBR Premix Ex Taq (Takara Biomedical) and the Thermal Cycler Dice Real-time System (Takara Biomedical). The PCR conditions for glyceraldehyde-3-phosphate dehydrogenase (GAPDH), anti-phospho-fibroblast growth factor receptor 1 (FGFR1), C-C chemokine receptor type 2 (CCR2), and anti-phospho- serum and glucocorticoid regulated kinase (SGK1) were: 94 °C for 2 min; followed by 40 cycles at 94 °C for 0.5 min, 50 °C for 0.5 min, and 72 °C for 0.5 min. The following primers were used: FGFR1, 5′- CTT CGT TTC TTG GTA TGC -3′ (5′-primer) and 5′- GGA CAG GAT GGA GTT TGG AC -3′ (3′-primer); CCR2, 5′- CTG TGT TTG CTT CTG TCC -3′ (5′-primer) and 5′- CCC TAT GCC TCT TCT C -3′ (3′-primer); SGK1, 5′- AGG ATG GGT CTG AAC GAC TTT -3′ (5′-primer) and 5′- GCC CTT TCC GAT CAC TTT CAA G -3′ (3′-primer); and GAPDH, 5′-ACT TTG TCA AGC TCA TTT-3′ (5′-primer) and 5′-TGC AGC GAA CTT TAT TG-3′ (3′-primer). The expression of FGFR1, CCR2, and SGK1 in KMS-26, KMS-28BM, and KMS-20 cells was analyzed as previously described [25,26,27].

### 2.6. Proteasome Activity Assay

Proteasome β5 subunit activity was assessed as described previously [28].

### 2.7. Autophagy Assay

Bortezomib- and ixazomib-induced autophagy was assessed using the Muse™ Cell Analyzer and Muse™ Autophagy LC3-antibody based Kit (Merck Millipore, Nottingham, UK). The autophagy induction ratios were analyzed for bortezomib- and ixazomib-treated cells versus control cells (0.1% dimethyl sulfoxide (DMSO)-treated cells).

### 2.8. Luminex Assay

Phosphorylated signaling protein expression was analyzed using a multiplex kit, Millipore Luminex 200^TM^ (Merck Millipore), using the following antibodies: anti-phospho-mammalian target of rapamycin (mTOR) (Ser2448), anti-phospho-Akt (Ser473), anti-phospho-c-Jun N-terminal kinase (JNK) (Thr183/Tyr185), anti-phospho-extracellular regulated protein kinase 1/2 (ERK1/2) (Thr185/Tyr187), anti-phospho-signal transducer and activator of transcription 3 (STAT3) (Tyr705), anti-phospho-p38 mitogen-activated protein kinase (p38MAPK) (Thr180/Tyr182), and anti-phospho-nuclear factor (NF)-κB (Ser536) (Merck Millipore).

### 2.9. Gene Expression Omnibus (GEO) Data Set

The gene expression profiles of the microarray datasets with the accession numbers GSE6205 and GSE9782 were obtained from the National Center of Biotechnology Information (NCBI) Gene Expression Omnibus (GEO) database (http://www.ncbi.nlm.nih.gov/geo/). The expression of various genes in KMS-26, KMS-28BM, and KMS-20 cells was analyzed from GSE6205 and the expression of anti-phospho- serum and glucocorticoid regulated kinase (SGK)1 and nuclear factor (NF)-κB p65, and the overall survival rate of patients with multiple myeloma (MM) were analyzed from GSE9782.

### 2.10. Statistical Analysis

All results are represented as the means and standard deviations (SDs) of several independent experiments. All analyses were conducted using SPSS version 21.0 software (IBM Inc., Chicago, IL, USA), and Shapiro-Wilk analysis and one-way analysis of variance (ANOVA) were performed. When no differences in the Shapiro-Wilk test and satisfactory differences in ANOVA were confirmed, the data from the control group and various drug-treated groups were compared and analyzed using Dunnett’s test. When our data did not show normal distribution, they were analyzed using the Kruskal-Wallis test followed by the Steel test. Survival rates were assessed using Kaplan-Meier curves and long-rank analysis. *p* values less than 0.05 were deemed significant. Drug interactions were analyzed using the combination index (CI) based on the method described by Chou and Talalay [29]. A CI value of less than 1.0 indicates synergy, while a CI value greater than 1 indicates antagonism.

## 3. Results

### 3.1. Sensitivity of Multiple Myeloma (MM) Cells to Bortezomib and Ixazomib

We investigated the cytotoxic effect of bortezomib (1–200 nM) and ixazomib (1–500 nM) on the KMS-20, KMS-26, and KMS-28BM cell lines. Although high concentrations of bortezomib (50–200 nM, *p* < 0.05) and ixazomib (100–500 nM, *p* < 0.05) induced KMS-20 cell death, low concentrations of bortezomib (5 nM, *p* < 0.05) and ixazomib (5 nM, *p* < 0.05) significantly induced KMS-26 and KMS-28BM cell death (Figure 1A,B). Additionally, KMS-20 cells showed a higher half-maximal inhibitory concentration (IC50) for bortezomib and ixazomib than KMS-26 and KMS-28BM cells (the IC50 value for bortezomib and ixazomib: KMS-20 vs. KMS-26 or KMS-28BM, *p* < 0.05) (Figure 1C). These results indicated that KMS-20 cells had a lower sensitivity to bortezomib and ixazomib than KMS-26 and KMS-28BM cells, and primary resistance to bortezomib and ixazomib.

### 3.2. Expression and Activity of Proteasome β5 Subunit and Effect of Autophagy on Bortezomib- and Ixazomib-Treated Multiple Myeloma (MM) Cells

Next, we examined the expression of the proteasome β5 subunit and the effect of bortezomib and ixazomib on proteasome β5 subunit activity and autophagy induction in the KMS-26, KMS-28BM, and KMS-20 cells. The expression level of the proteasome β5 subunit did not differ among the cell lines, and a similar concentration of bortezomib and ixazomib inhibited proteasome β5 subunit activity in KMS-26, KMS-28BM, and KMS-20 cells (*p* < 0.05) (Figure 2A–C). Treatment with bortezomib or ixazomib did not affect autophagy induction in the KMS-26, KMS-28BM, and KMS-20 cells (Figure 2D,E).

### 3.3. Overexpression of Anti-Phospho-Serum and Glucocorticoid-Regulated Kinase (SGK1) Correlated with Low Sensitivity of Bortezomib and Ixazomib in Multiple Myeloma (MM) Cells

To reveal differences in gene expression between KMS-20 cells and KMS-26 or KMS-28BM cells, we used the publicly available gene expression profiling (GEP) dataset, GSE6205. These analyses revealed that the expression of several genes was elevated only in KMS-20 cells. Above all, we focused on three genes; anti-phospho-fibroblast growth factor receptor 1 (FGFR1), a member of the growth factor receptor tyrosine kinase family; C-C chemokine receptor type 2 (CCR2); and anti-phospho- serum and glucocorticoid regulated kinase (SGK1), a member of the serine/threonine kinase family (Table 1). The FGFR family includes FGFR1, FGFR2, FGFR3, and FGFR4, and binding of FGFR to fibroblast growth factor (FGF) induces the activation of signaling molecules, such as RAS/ERK, phosphoinositide 3-kinase (PI3K)/Akt, Janus kinase (JAK)/signal transducer and activator of transcription (STAT), and the phospho- anti-phospho-c-Jun N-terminal kinase (JNK) pathway [30]. Additionally, activation of FGFR by FGF2 stimulation was involved in prednisolone resistance in B cell precursor acute lymphoblastic leukemia cells [31]. It has been reported that FGFR3 overexpression is less sensitive to bortezomib in U266 cells [32]. The CC chemokine ligand (CCL2)-CCR2 axis is involved in multi-tyrosine kinase inhibition and microtubule inhibition resistance in the colon and prostate cancer cells [33,34], and CCR2 promotes cell growth and cell cycle progression via Src and Akt activation in breast cancer cells [35]. Activation of SGK1 by tongue cancer resistance-related protein 1 induces tamoxifen resistance [36], and a high expression of SGK1 correlates with a poor prognosis in patients treated with neoadjuvant chemotherapy and esophageal squamous cell carcinoma [37]. These findings suggest that overexpression of FGFR1, CCR2, or SGK1 is involved in resistance to various anti-cancer drugs. Thus, the overexpression of FGFR1, CCR2, and SGK1 may explain the low sensitivity of multiple myeloma (MM) cells to bortezomib and ixazomib.

We investigated the expression of SGK1, CCR2, and FGFR1 mRNA and protein in KMS-26, KMS-28BM, and KMS-20 cells. The expression of SGK1, CCR2, and FGFR1 mRNA and total protein levels were elevated in KMS-20 cells compared to those in KMS-26 and KMS-28BM cells (Figure 3A,B). Additionally, phosphorylated SGK1 and FGFR1 levels were increased in KMS-20 cells with an increase in total SGK1 and FGFR1 protein levels (Figure 3B). Next, we examined whether the inhibition of these signal molecules by selective inhibitors enhanced the cytotoxicity of bortezomib and ixazomib in KMS-20 cells. PD166866, an FGFR1 inhibitor, and a CCR2 antagonist did not affect cell death induced by bortezomib and ixazomib but GSK650394, an SGK1 inhibitor, enhanced the cytotoxicity of bortezomib and ixazomib to KMS-20 cells (*p* < 0.05) (Figure 3C–E). Additionally, combined treatment with GSK650394 and bortezomib or ixazomib enhanced the Annexin V-positive cells compared to treatment with bortezomib, ixazomib, and GSK650394 alone (Appendix A). Moreover, bortezomib and ixazomib enhanced the SGK1 total or phosphorylated protein in KMS-20 cells (Appendix A). These results indicate that bortezomib and ixazomib enhance SGK1 overexpression/activation and that the SGK1 inhibitor overcame bortezomib- and ixazomib-resistance via suppression of SGK1 activation in KMS-20 cells.

To analyze whether the expression of SGK1 is correlated with the response to bortezomib and overall survival in patients with MM, we used the publicly available GEP dataset, GSE9782. Patients with high SGK1 expression had a lower sensitivity to bortezomib and a shorter overall survival than patients with low SGK1 expression (*p* < 0.01) (Figure 3F,G). Thus, the overexpression and activation of SGK1 were associated with the low sensitivity of MM cells to bortezomib and ixazomib.

### 3.4. GSK650394 Inhibited Nuclear Factor (NF)-κB Activation and Regulated the Expression of Cell Survival-Related Molecules

It has been indicated that activation of serum/glucocorticoid regulated kinase 1 (SGK1) promotes nuclear factor (NF)-κB and phospho-mammalian target of rapamycin (mTOR) activation and suppresses phospho- anti-phospho-c-Jun N-terminal kinase (JNK) activation [38,39,40]. It has been reported that activation of the JAK/STAT3, mitogen-activated protein kinase kinase (MEK)/ERK, PI3K/Akt, or phospho-p38 mitogen-activated protein kinase (p38MAPK) pathways induces the transcription of SGK1 [19,41,42]. Therefore, we examined the activation of SGK1 downstream signaling molecules and the induction of SGK1 expression. Regarding KMS-20 cells, the activation of Akt, JNK, ERK1/2, and NF-κB p65 was higher than that in KMS-26 and KMS-28BM cells (*p* < 0.01) (Figure 4A,B). The activation of mTOR was higher in KMS-20 and KMS-28BM cells than in KMS-26 cells (*p* < 0.01) (Figure 4A). The activation of STAT3 and p38MAPK was similar among these cells (Figure 4A). Next, we investigated whether GSK650394 suppresses ERK1/2, NF-κB p65, Akt, and JNK activation via suppression of SGK1 activation in KMS-20 cells. GSK650394 suppressed NF-κB p65 activation and nuclear translocation of NF-κB p65, but not ERK1/2, Akt, or JNK activation in KMS-20 cells (Figure 4C). Additionally, we found that U126, a mitogen-activated protein kinase kinase 1/2 (MEK1/2) inhibitor; LY924002, a PI3K inhibitor; and SP600125, a JNK inhibitor, did not suppress SGK1 expression, nor did it affect bortezomib and ixazomib sensitivity in KMS-20 cells (Appendix A). Recently, it was reported that SGK1 inhibition enhanced the cytotoxicity of bortezomib via activation of the JNK pathway and induction of CCAAT/enhancer binding protein (C/EBP) homologous protein (CHOP) expression, an essential factor of ER stress, in bortezomib-resistant MM cells [20]. Therefore, we examined whether the SGK inhibitor enhanced CHOP expression in KMS-20 cells; however, we did not observe enhanced CHOP expression in KMS-20 cells upon SGK1 inhibition (Appendix A). It has been reported that activation of NF-κB regulates apoptosis-regulated factors, such as the anti-B-cell lymphoma-2 (Bcl-2) family and inhibitor of apoptosis protein (IAP) family [43,44]. We found that GSK650394 suppressed Bcl-2, Bcl-xL, and survivin expression, enhanced anti-Bcl-2-like protein 11 (Bim) expression, and did not affect anti-Bcl-2-associated X (Bax), anti-X-linked inhibitor of apoptosis protein (XIAP), anti-phorbol-12-myristate-13-acetate-induced protein (Noxa), and anti-p53 upregulated modulator of apoptosis (Puma) expression in KMS-20 cells (Figure 4D). Moreover, SGK1 short interfering RNA (siRNA) suppressed the expression of phosphorylated NF-κB p65 and nuclear translocation of NF-κB p65 and enhanced the sensitivity to bortezomib and ixazomib in KMS-20 cells with similar results to those for GSK650394 (Appendix A). These results suggest that SGK1 promoted the expression of phosphorylated NF-κB p65 and the nuclear translocation of NF-κB p65, and the SGK1 inhibitor suppressed the Bcl-2, Bcl-xL, and Survivin expression, and enhanced the Bim expression via inhibition of the SGK1/NF-κB pathway.

Moreover, we analyzed whether the expression of NF-κB p65 is associated with the response to bortezomib and the overall survival in patients with multiple myeloma (MM) using the GSE9782 dataset. High expression of NF-κB p65 in patients with MM led to a low sensitivity to bortezomib and a shorter overall survival than low expression of NF-κB p65 in patients (*p* < 0.01) (Figure 4E,F). Thus, both the overexpression and activation of NF-κB p65 were associated with the low sensitivity of MM cells to bortezomib and ixazomib.

### 3.5. Dimethyl Fumarate (DMF), a Nuclear Factor (NF)-κB Inhibitor, Enhanced the Sensitivity of KMS-20 Cells to Bortezomib and Ixazomib

Our results suggest that a low sensitivity to bortezomib and ixazomib is correlated with the activation of nuclear factor (NF)-κB by serum/glucocorticoid regulated kinase 1 (SGK1). We examined whether dimethyl fumarate (DMF), an NF-κB inhibitor, enhances the sensitivity of KMS-20 cells to bortezomib and ixazomib. The combination of DMF and bortezomib or ixazomib enhanced the sensitivity of KMS-20 cells to bortezomib and ixazomib (*p* < 0.05) (Figure 5A,B and Appendix A). Additionally, DMF suppressed B-cell lymphoma-2 (Bcl-2), Bcl-xL, and survivin expression and enhanced Bcl-2-like protein 11 (Bim) expression via the inhibition of NF-κB nuclear localization in KMS-20 cells (Figure 5).

To validate these observations, we confirmed the sensitivity to bortezomib and ixazomib in other multiple myeloma (MM) cells. We found that L363 had a lower sensitivity to bortezomib and ixazomib, similar to that of KMS-20 cells, but ARH-77 and RPMI8226 cells showed a high sensitivity to bortezomib and ixazomib, which was similar to that of the KMS-26 and KMS-28BM cells (Appendix A). Additionally, activation of SGK1 and NF-κB p65 was higher in the L363 cells than in the ARH-77 and RPMI8226 cells, and GSK650394 and DMF enhanced the sensitivity to bortezomib and ixazomib in L363 cells (Appendix A). These results indicate that the SGK/NF-κB pathway may play a role in the underlying mechanism of the low sensitivity of KMS-20 and L363 cells to bortezomib and ixazomib.

## 4. Discussion

Here, KMS-20 cells presented a lower sensitivity to bortezomib and ixazomib than KMS-26 and KMS-28BM cells. Although proteasome β5 subunit overexpression and the induction of autophagy by a proteasome inhibitor have been associated with resistance in multiple myeloma (MM) cells [45,46,47], our results showed that there was no change in the expression level of the proteasome β5 subunit, inhibition level of proteasome β5 subunit activity, or degree of autophagy induction by bortezomib and ixazomib. These results suggest that a low sensitivity to bortezomib and ixazomib relies on other factors.

To ascertain the mechanism underlying the low sensitivity to bortezomib and ixazomib, we analyzed the gene expression profiling (GEP) database to reveal the genes expressed in KMS-26 or KMS-28BM cells, compared to those in KMS-20 cells. Considering the possible resistance genes revealed in this study, fibroblast growth factor receptor 1 (FGFR1), C-C chemokine receptor type 2 (CCR2), and serum/glucocorticoid regulated kinase 1 (SGK1) were overexpressed in KMS-20 cells. FGFR1 contributes to tyrosine kinase inhibitor resistance and chemoresistance in lung, breast, and urothelial cancers [48,49,50]. The activation of CCR2 is involved in resistance to regorafenib, a multikinase inhibitor, and cabazitaxel in colon and prostate cancer cells [33,34]. SGK1 is associated with Akt and phosphoinositide 3-kinase inhibitor and paclitaxel resistance in breast and ovarian cancer cells [51,52,53]. Additionally, SGK1, CCR2, and SGK1 mRNA and protein expression, and the activation of SGK1 and FGFR1, were higher in KMS-20 cells than in KMS-26 and KMS-28BM cells. Furthermore, the inhibition of SGK1 by an SGK1 inhibitor, GSK650394, enhanced the cytotoxic effect of bortezomib and ixazomib, whereas the combined administration of bortezomib or ixazomib with an FGFR1 inhibitor, PD166866, or a CCR2 antagonist did not. Moreover, patients with MM who were non-responsive to bortezomib had a high expression of SGK1 and those with a high expression of SGK1 had a significantly lower overall survival than patients with a low expression of SGK1. Thus, the overexpression and activation of SGK1 have important roles in a low sensitivity to bortezomib and ixazomib.

Although it is known that the activation of SGK1 accelerates the phosphorylation of N-myc downstream regulated gene 1 and activation of murine double minute 2, the effects of signaling crosstalk is unknown [54]. The expression of extracellular regulated protein kinase 1/2 (ERK1/2), nuclear factor (NF)-κB, Akt, and c-Jun N-terminal kinase (JNK) was higher in KMS-20 cells than in KMS-26 and KMS-28BM cells. GSK650394 suppressed phosphorylated NF-κB expression but did not affect ERK1/2, Akt, and JNK phosphorylation. Moreover, GSK650394 inhibited B-cell lymphoma-2 (Bcl-2), Bcl-xL, and survivin expression, increased Bcl-2-like protein 11 (Bim) expression and did not affect Bcl-2-associated X (Bax), X-linked inhibitor of apoptosis protein (XIAP), phorbol-12-myristate-13-acetate-induced protein 1 (Noxa), and p53 upregulated modulator of apoptosis (Puma) expression. The combination of dimethyl fumarate (DMF), an NF-κB inhibitor, and bortezomib or ixazomib strongly induced cell death in KMS-20 cells via the suppression of NF-κB activation, which inhibited the expression of Bcl-2, Bcl-xL, and survivin and enhanced the expression of Bim. NF-κB p65 phosphorylation at Ser536 is involved in transcriptional activity, and NF-κB p65 activation modulates the expression of several apoptosis-regulating factors, such as those in the Bcl-2 and IAP families [55,56,57,58,59,60]. Activation of NF-κB promotes the expression of Bc-2, Bcl-xL, survivin, and XIAP, and downregulated Puma, Noxa, Bim, and Bax expression via suppression of p53 function [43,44,61,62,63]. Bcl-2 and Bcl-xL suppress the apoptosis-inducing function of proapoptotic proteins such as Bax, Bim, Noxa, and Puma [64]. Survivin and XIAP inhibit the function of caspases, which are apoptosis-leading proteinases, and suppress the induction of apoptosis [65]. The Bcl-2 overexpression was associated with a low clinical response to bortezomib in MM patients [66]. Moreover, it indicated that upregulation of Bcl-xL by NF-κB signaling activation induced proteasome inhibitor resistance in MM cells [67]. Additionally, low levels of Bim expression were associated with bortezomib resistance in bortezomib-resistant U266PS-R cells and primary MM cells, whereas ABT-737, a Bcl-2 homology 3 (BH3) mimetic, overcame the bortezomib resistance in U266PS-R cells [68]. Furthermore, Survivin was overexpressed in the bortezomib-resistant NCI-H929-R20.1 cells compared to the parent cells, and BAY-11-7082, an IκB kinase inhibitor, suppressed the Survivin expression via inhibition of NF-κB p65 activation in bortezomib-resistant NCI-H929-R20.1 cells [60]. Moreover, NF-κB p65 activation induced bortezomib resistance, but the inhibition of NF-κB signaling reduced bortezomib resistance in MM cells [69,70]. Moreover, we found that high expression of NF-κB p65 in MM patients was associated with a low sensitivity to bortezomib and a poor prognosis compared to that of patients with a low NF-κB p65 expression. These results suggest that the SGK/NF-κB signaling pathway contributes to a low sensitivity to bortezomib and ixazomib and an SGK or NF-κB inhibitor may enhance the cytotoxic effect of bortezomib and ixazomib in MM cells.

It has been demonstrated previously that SGK1 activation is involved in acquired bortezomib resistance in MM cells. Also, SGK1 inhibition by bortezomib or short hairpin RNA (shRNA) has been shown to enhance its cytotoxicity via activation of the JNK pathway and induction of endoplasmic reticulum (ER) stress [20]. During the present study, we showed that the overexpression of SGK1 is correlated with primary resistance to bortezomib and ixazomib via activation of the NF-κB pathway. Moreover, the inhibition of SGK1 and NF-κB by the inhibitors, bortezomib and ixazomib, aided in repressing primary resistance. Additionally, we found that SGK1 inhibitors did not induce CHOP expression and activation of JNK. These findings indicate that the activation of the SGK1/NF-κB pathway plays an important role in developing bortezomib and ixazomib resistance in MM cells.

It has previously been indicated that induction of SGK1 expression was regulated via activation of the JAK/STAT3, MEK/ERK, PI3K/Akt, or p38MAPK pathways [19,41,42]. However, in this study we found that the extent of activation of the STAT3 and p38MAPK pathways was comparable among the KMS-20, KMS-26, and KMS-28 cells. Additionally, U0126, a MEK inhibitor, and LY294002, a PI3K inhibitor, did not affect the expression of the SGK1 protein, and did not enhance the cytotoxicity of bortezomib and ixazomib in KMS-20 cells. These results indicate that SGK1 transcription is not involved in the activation of the JAK/STAT3, MEK/ERK, PI3K/Akt, or p38MAPK pathways in KMS-20 cells. However, the mechanism underlying SGK1 induction via these signaling pathways is not yet clear. Therefore, future studies are required to investigate their effects on SGK1 transcription.

## 5. Conclusions

To conclude, both the overexpression and activation of the serum/glucocorticoid regulated kinase 1 (SGK1)/Nuclear Factor (NF)-κB pathway are involved with the low sensitivity of multiple myeloma (MM) cells to bortezomib and ixazomib, and SGK1 or NF-κB inhibitors can be used to increase the cytotoxic effects of bortezomib and ixazomib. These results indicate that the combination of an SGK1 or NF-κB inhibitor and bortezomib or ixazomib may be a potential therapy for MM harboring the overexpression and activation of the SGK1/NF-κB pathway.

## Figures and Tables

**Figure 1 biomedicines-09-00033-f001:**
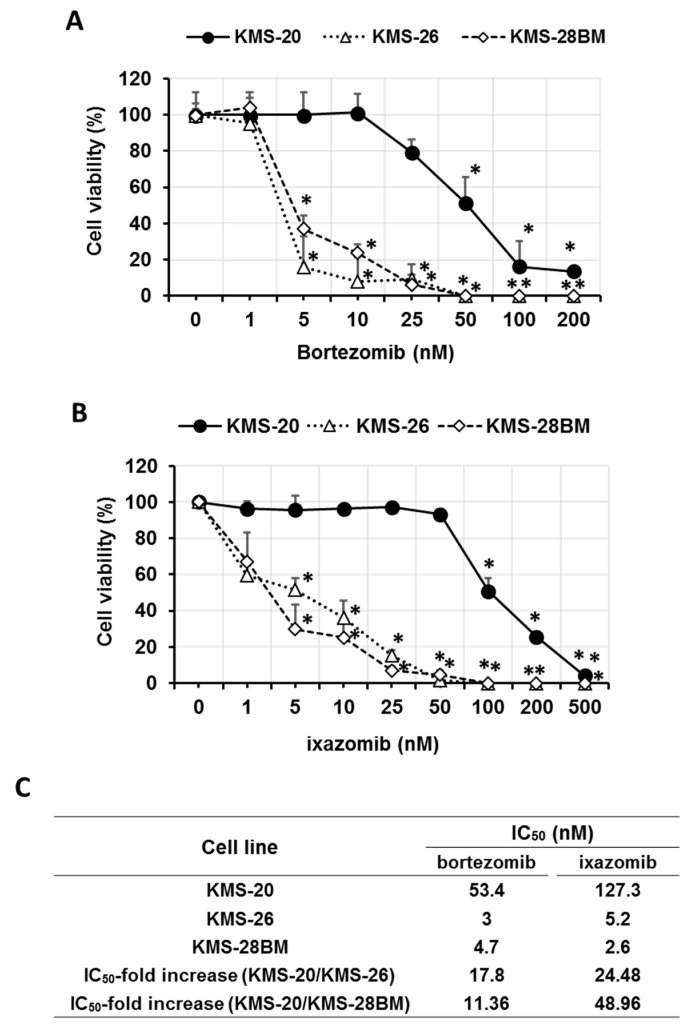
Effect of bortezomib and ixazomib on human multiple myeloma cell viability. Viability of (**A**) bortezomib- and (**B**) ixazomib-treated KMS-20, KMS-26, and KMS-28BM cells, as measured by the trypan blue dye assay. These cells were treated with the indicated concentrations of bortezomib for 3 days. The results are representative of five independent experiments. * *p* < 0.01 vs. controls (viability of KMS-20 cells was analyzed by the Shapiro-Wilk test and one-way analysis of variance (ANOVA) with Dunnett’s test. The viability of the KMS-26 and KMS-28BM cells was analyzed by the Shapiro-Wilk and Kruskal-Wallis tests, followed by the Steel test). (**C**) half-maximal inhibitory concentration (IC50) of bortezomib and ixazomib for KMS-20, KMS-26, and KMS-28BM cells.

**Figure 2 biomedicines-09-00033-f002:**
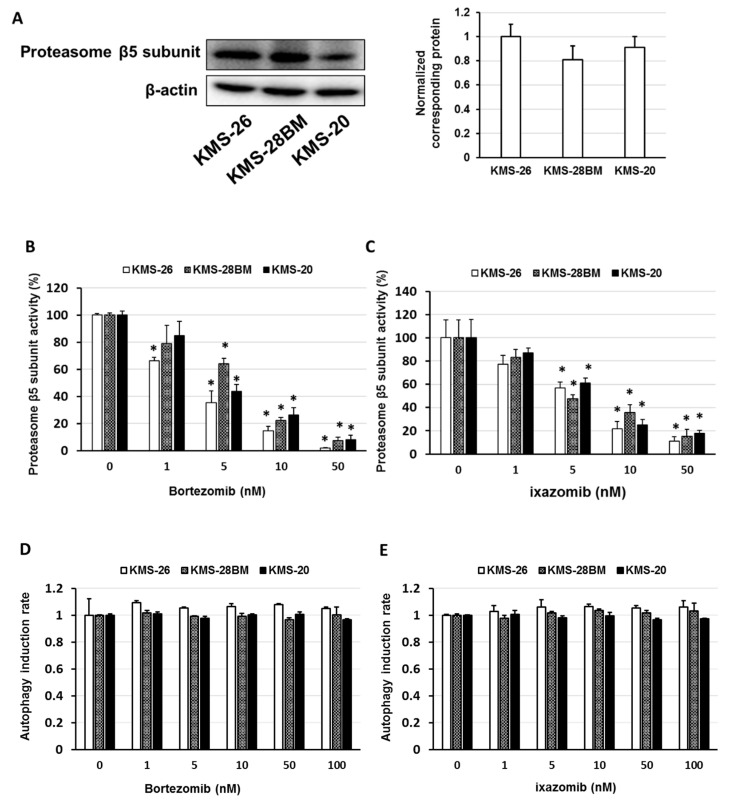
Effect of bortezomib and ixazomib on proteasome β5 subunit activity and autophagy induction. (**A**) Cell lysates were examined by western blotting using the indicated antibodies. Quantification of the amount of proteasome β5 subunit, normalized to the amounts of β-actin. The results are representative of three independent experiments. (**B**,**C**) Cells were treated with (**B**) bortezomib or (**C**) ixazomib for 8 hr. Control cells (0 μM) were administered 0.5% dimethyl sulfoxide (DMSO) for 8 hr. Cell extracts were incubated for 1.5 h, at which point the fluorogenic peptide substrate 7-amino-4-methylcoumarin, which detects proteasome β5 subunit activity, was added to the extracts. The fluorescence assays (excitation, 360 nm; emission, 465 nm) were conducted at room temperature. These results are representative of five independent experiments. * *p* < 0.01 vs. controls (viability of KMS-20 cells was analyzed by the Shapiro-Wilk test and one-way analysis of variance (ANOVA) with Dunnett’s test. (**D**,**E**) Cells were treated with (**D**) bortezomib or (**E**) ixazomib for 1 day. Control cells (0 μM) were administered 0.5% DMSO for 1 day. Autophagy induction was analyzed using the Muse™ Cell Analyzer and Muse™ Autophagy LC3-antibody-based kit.

**Figure 3 biomedicines-09-00033-f003:**
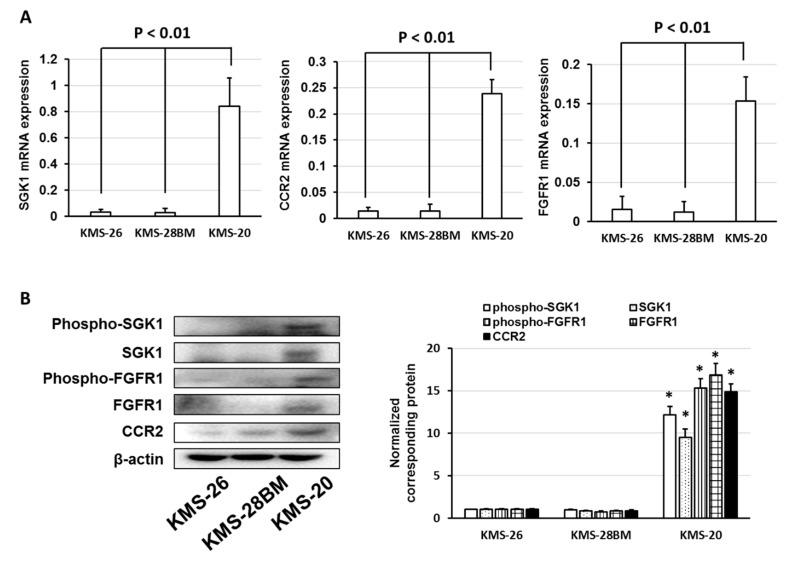
Serum/glucocorticoid regulated kinase 1 (SGK1) inhibitor enhanced the cytotoxic effects of bortezomib and ixazomib. (**A**) Expression of SGK1, C-C chemokine receptor type 2 (CCR2), and fibroblast growth factor receptor 1 (FGFR1) in KMS-26, KMS-28BM, and KMS-20 cells. Total RNA was extracted and SGK1, CCR2, and FGFR1 levels were determined by real-time polymerase chain reaction (PCR). The results are expressed as the test:control ratio after normalization using glyceraldehyde-3-phosphate dehydrogenase (GAPDH). The results are representative of five independent experiments. * *p* < 0.01 vs. KMS-26 or KMS-28BM cells as assessed by Dunnett’s test. (**B**) Cell lysates were examined by western blotting using the indicated antibodies. Quantification of the amount of phospho-SGK1, SGK1, phospho-FGFR1, FGFR1, and CCR2, normalized to the amounts of β-actin. The results are representative of three independent experiments. * *p* < 0.01, compared to controls (one-way analysis of variance ANOVA with Dunnett’s test). (**C**–**E**) KMS-20 cells were administered the indicated concentrations of bortezomib, ixazomib, PD166866, CCR2 antagonist, or GSK650394. After incubation for 72 h, cell viability was analyzed by trypan blue staining. The combination index (CI) values of bortezomib, ixazomib, and GSK650394 for KMS-20 cells were noted. The results are representative of five independent experiments. * *p* < 0.01 vs. untreated cells, as assessed with the Shapiro-Wilk test and one-way analysis of variance (ANOVA) with Dunnett’s test. (**F**) Expression of SGK1 in bortezomib-non-responders and bortezomib-responders was analyzed using the GSE9782 dataset. (**G**) Correlation between SGK1 expression and multiple myeloma prognosis (overall survival). Survival rates were assessed using Kaplan-Meier curves and long-rank analysis.

**Figure 4 biomedicines-09-00033-f004:**
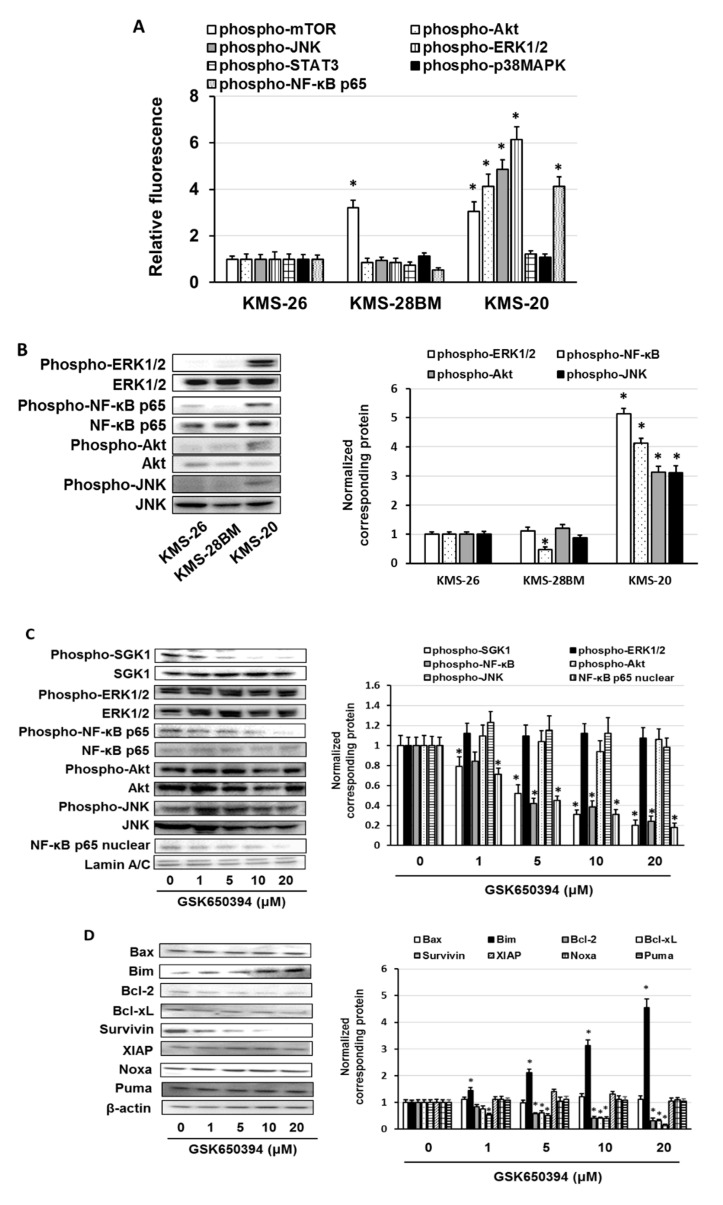
Effect of GSK650394 on extracellular regulated protein kinase 1/2 (ERK1/2), nuclear factor-κB (NF-κB), Akt, and c-Jun N-terminal kinase (JNK) activation and expression of Bcl-2-associated X (Bax), Bcl-2-like protein 11 (Bim), B-cell lymphoma-2 (Bcl-2), B-cell lymphoma-extra large (Bcl-xL), survivin, X-linked inhibitor of apoptosis protein (XIAP), phorbol-12-myristate-13-acetate-induced protein 1 (Noxa), and p53 upregulated modulator of apoptosis (Puma). (**A**) Cells were lysed and the phosphorylation of the mammalian target of rapamycin (mTOR), Akt, JNK, ERK1/2, the signal transducer and activator of transcription 3 (STAT3), p38 mitogen-activated protein kinase (p38MAPK), and NF-κB was measured by the Luminex assay. (**B**) Cell lysates were examined by western blotting using the indicated antibodies. Quantification of the amount of phospho-ERK1/2, phospho-NF-κB p65, phospho-Akt, or phospho-JNK, normalized to the amounts of ERK1/2, NF-κB, Akt, or JNK. The results are representative of three independent experiments. * *p* < 0.01, compared to controls (one-way analysis of variance ANOVA with Dunnett’s test). (**C**,**D**) Cells were treated with GSK650394 for 3 days. Control cells (0 μM) were treated with 0.5% dimethyl sulfoxide (DMSO) for 3 days. Cell lysates were examined by western blotting using the indicated antibodies. Quantification of the amount of phospho- serum/glucocorticoid regulated kinase 1 (SGK1), phospho-ERK1/2, phospho-NF-κB p65, phospho-Akt, phospho-JNK, Bax, Bim, Bcl-2, Bcl-xL, Survivin, XIAP, Noxa, or Puma, normalized to the amounts of SGK1, ERK1/2, NF-κB, Akt, JNK, or β-actin. The results are representative of three independent experiments. * *p* < 0.01, compared to controls (one-way ANOVA with Dunnett’s test). (**E**) Expression of NF-κB p65 in bortezomib-non-responders and bortezomib-responders was analyzed using the GSE9782 dataset. (**F**) Correlation between NF-κB p65 expression and multiple myeloma prognosis (overall survival). Survival rates were assessed using Kaplan-Meier curves and long-rank analysis.

**Figure 5 biomedicines-09-00033-f005:**
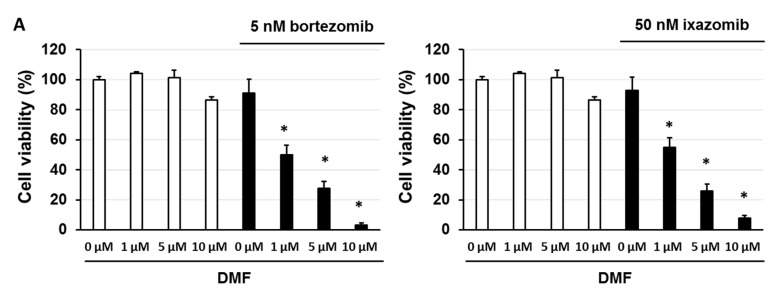
Nuclear factor-κB (NF-κB) inhibitor enhanced the cytotoxic effects of bortezomib and ixazomib. (**A**) KMS-20 cells were administered the indicated concentrations of bortezomib, ixazomib, or dimethyl fumarate (DMF). After incubation for 72 h, cell viability was analyzed by trypan blue staining. The results are representative of five independent experiments. * *p* < 0.01 vs. untreated cells, as assessed by the Shapiro-Wilk test and one-way analysis of variance (ANOVA) with Dunnett’s test. (**B**) Cells were treated with DMF for 3 days. Control cells (0 μM) were treated with 0.5% dimethyl sulfoxide (DMSO) for 3 days. The cell lysates were examined by western blotting using the indicated antibodies. Quantification of the amount of NF-κB p65 nuclear, NF-κB p65 cytoplasm, Bcl-2-associated X (Bax), Bcl-2-like protein 11 (Bim), B-cell lymphoma-2 (Bcl-2), Bcl-xL, Survivin, X-linked inhibitor of apoptosis protein (XIAP), phorbol-12-myristate-13-acetate-induced protein 1 (Noxa), or p53 upregulated modulator of apoptosis (Puma), normalized to the amounts of β-actin or Lamin A/C. The results are representative of three independent experiments. * *p* < 0.01, compared to controls (one-way ANOVA with Dunnett’s test).

**Table 1 biomedicines-09-00033-t001:** Identification of overexpressed genes in KMS-20 cells compared to those in KMS-26 and KMS-28BM cells using the GSE6205 dataset.

Gene Symbol	Gene Name	logFC	*p* Value
CFH	complement factor H	6.16	0.0000457
CRIP1	Cysteine-rich protein 1	5.71	0.0000437
GPX1	glutathione peroxidase 1	4.56	0.0001157
VCX2	variable charge, X-linked 2	4.51	0.0001059
CCR2	C-C motif chemokine receptor 2	4.5	0.0001194
VCX2	variable charge, X-linked 2	4.24	0.0001347
PHLDA2	pleckstrin homology-like domain family A member 2	4.21	0.0002172
CCR2	C-C motif chemokine receptor 2	4.14	0.0001985
SGK1	serum/glucocorticoid regulated kinase 1	4.08	0.0001536
CD52	CD52 molecule	4.08	0.0001583
CALML4	Calmodulin-like 4	4.08	0.0005235
S100A4	S100 calcium binding protein A4	3.99	0.0011388
COL21A1	collagen type XXI alpha 1 chain	3.92	0.0002591
IFITM2	Interferon-induced transmembrane protein 2	3.91	0.0004026
SCN3A	sodium voltage-gated channel alpha subunit 3	3.86	0.0003191
BHLHE41	basic helix-loop-helix family member e41	3.78	0.0012547
SORL1	Sortilin-related receptor 1	3.75	0.0002626
MOXD1	monooxygenase DBH-like 1	3.74	0.0011837
CALML4	Calmodulin-like 4	3.68	0.0007518
EPCAM	epithelial cell adhesion molecule	3.64	0.0003476
MEST	Mesoderm-specific transcript	3.62	0.0002583
SLC25A21	solute carrier family 25 member 21	3.58	0.0002612
ARMCX2	armadillo repeat containing, X-linked 2	3.56	0.0003608
PLS3	plastin 3	3.49	0.0003084
NLGN4X	neuroligin 4, X-linked	3.47	0.0003288
RFTN1	raftlin, lipid raft linker 1	3.46	0.0004193
IFITM1	Interferon-induced transmembrane protein 1	3.46	0.0005397
ACTN1	actinin alpha 1	3.44	0.0028258
TUSC3	tumor suppressor candidate 3	3.42	0.0003053
GAGE3	G antigen 3	3.41	0.0004645
RNASET2	ribonuclease T2	3.41	0.0006196
THY1	Thy-1 cell surface antigen	3.39	0.0003143
PTN	pleiotrophin	3.37	0.0003203
CD37	CD37 molecule	3.3	0.0013067
HHLA2	HERV-H LTR-associating 2	3.29	0.0004471
RNASET2	ribonuclease T2	3.09	0.0004523
SLC2A3	solute carrier family 2 member 3	3.05	0.0011833
PLAC1	placenta specific 1	2.98	0.0013038
FGFR1	fibroblast growth factor receptor 1	2.95	0.0005482
KYNU	kynureninase	2.92	0.0006091
FCRL2	Fc receptor-like 2	2.9	0.0006978
NPDC1	neural proliferation, differentiation, and control 1	2.84	0.0015714
SGPP1	sphingosine-1-phosphate phosphatase 1	2.84	0.0019385
PSTPIP1	proline-serine-threonine phosphatase interacting protein 1	2.81	0.0015156
NLRP2	NLR family pyrin domain containing 2	2.78	0.00072
ANXA3	annexin A3	2.74	0.0007048
IDUA	iduronidase, alpha-L-	2.74	0.0030465
SPARC	secreted protein acidic and cysteine-rich	2.72	0.0007207
TSPYL5	TSPY-like 5	2.71	0.0007929
SORL1	Sortilin-related receptor 1	2.7	0.0012272
SEMA4A	semaphorin 4A	2.68	0.0009015
TUSC3	tumor suppressor candidate 3	2.67	0.0007865
LCP2	lymphocyte cytosolic protein 2	2.63	0.0009132
FMO3	Flavin-containing monooxygenase 3	2.61	0.000845
TPTE	transmembrane phosphatase with tensin homology	2.61	0.0010097
ABCC1	ATP-binding cassette subfamily C member 1	2.58	0.0013177
TNIK	TRAF2- and NCK-interacting kinase	2.53	0.0009599
THY1	Thy-1 cell surface antigen	2.52	0.0010731
SERPINE2	serpin family E member 2	2.52	0.0027214
BASP1	brain abundant membrane attached signal protein 1	2.52	0.0049738
ALOX5AP	arachidonate 5-lipoxygenase-activating protein	2.51	0.0049432
C1orf54	chromosome 1 open reading frame 54	2.5	0.0013496

## Data Availability

The data presented in this study are availabl on request from the corresponding author.

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
