# Peer review of "Activation of Serum/Glucocorticoid Regulated Kinase 1/Nuclear Factor-κB Pathway Are Correlated with Low Sensitivity to Bortezomib and Ixazomib in Resistant Multiple Myeloma Cells"

_biomedicines, 2021, doi:10.3390/biomedicines9010033_

Round 1

Reviewer 1 Report

In this manuscript by Tsubaki et al, authors used a set of MM cell lines with different degrees of sensitivity/resistance to first and second-generation proteasome inhibitors (PIs), namely bortezomib and ixazomib, to identify new mechanism of resistance to these drugs. Using publicly available database, they selected three potential markers of PI refractoriness and by pharmacological inhibition, coupled to cell viability assays and western blot analysis, they identified constitutive serum/glucocorticoid regulated kinase 1 (SGK1) activation and downstream NF-kB signaling as a mechanisms potentially involved in the resistance phenotype.

Despite an interesting initial hypothesis, several inconsistencies were detected through the experimental design of this study. Together with a limited set of cell lines with no validation experiments in MM primary cultures and/or MM xenograft models, these data should be considered rather preliminary.

Major issues:

  • Based on the gene list shown on Table 1, authors arbitrarily choose to validate CCR2 (#5), SGK1 (#9) and FGFR1 (#39) which are far from being the most differentially-expressed genes between PI-sensitive and PI-resistant MM cell lines. No methodological, bibliographic or biological rationales are provided to support this choice.
  • Figures 1B-C: inhibition of proteasome activity is usually evaluated at short time (below 10 hours) after exposure to PI (PMID: 16140960). The time point used there (24 hours) is not suitable for such an analysis.
  • Figure 3B: authors report improperly an “activation” SGK1 and FGFR1 following western blot examination (line 193), whereas increased p-SGK1 and p-FGFR1 levels are likely due to an increase of total protein levels for both factors, in KMS-20 cells.
  • Figure 3E: this kind of combination study should have been accompanied by a drug interaction analysis using Chou and Talalay’s algorithm, and the corresponding Combination Indexes should have been provided. In addition, a dose-dependent evaluation of p-SGK1 protein levels upon by bortezomib/ixazomib should accompanied the drug interaction study to support a synergistic modulation of p-SGK1.
  • Line 205: in the absence of functional experiment, at this stage the term “involved” should be changed to “associated”
  • Figures 4A: basal protein expression WB only indicates the presence of higher phospho-kinase levels in KMS-20 cells in association with the higher SGK1 expression shown in Figure 3B. These sole results do not support the claim of the authors that “SGK1 had the potential to the activation of ERK1/2, NF-κB p65, Akt, and JNK” (line 231). Actually, Figure 4C only confirms the potential correlation between p-SGK1 inhibition and p-p65 NF-kB decrease. Importantly, in this last figure phospho- and total SGK1 levels should be examined to validate the activity of GSK650394 and densitometric quantification should be provided.
  • Figure 4D: authors omitted to discuss the link between inhibition of SGK1, NFkB signaling blockade and modulation of Bcl-2 family of proteins.
  • Fig S3A: SGK1 knockdown efficacy should be quantified by densitometric analysis. Downregulation seems to be reduce compared to the important decrease in NFkB activation. How can the authors explain such a difference between both factors?

Minor issues:

  • Abstract and introduction: distinguish between “intractable” and “incurable”

FigS6 E-F legend indicates KMS20 while this figure refers to L363

Reviewer 2 Report

Dear. Dr. Nishida

I have reviewed the manuscript entitled “Activation of serum/glucocorticoid regulated kinase 1/nuclear factor-κB pathway are correlated with low sensitivity to bortezomib and ixazomib in resistant multiple myeloma cells”. In this manuscript, you analyzed the molecules which correlated with bortezomib and ixazomib resistance using some cell lines. Although the concept proposed by the authors is interesting and your findings might lead to a novel therapy in the future, there are various serious analytical and technical issues supporting their claim, and the manuscript is partially not written well.

Major comments.

  1. As you referred in your manuscript (ref. 26), there is a murine xenograft model with SGK over-expression. If you want to insist on your point, you should test the effects of SGK1/NF-κB on resistance to bortezomib and ixazomib using in vivo model.

  1. I think some information of SGK1 and its general functions, furthermore previous reports about SGK1 on MM are needed in introduction.

    3. The authors focused on three genes, FGFR1, CCR2, and SGK1,           because these genes were highly expressed in KMS-20 cells                 compared to other two cell lines. However, the data in Table 1             suggests that many genes other than FGFR1, CCR2, and SGK1             are highly expressed in KMS-20 cells compared to other two cell           lines. The authors should clarify these discrepancies to justify               their claim.

  1. You analyzed several survival-related molecules showed in Figure 4. You mentioned these molecules are SGK1 downstream signaling molecules (line 225). Is that correct? As you described, Akt and associated mTOR, and NF-κB are SGK1 downstream signaling molecules. But other than them, they are different pathways. You should describe them accurately.

Minor Comments

  1. The content of abstract is inadequate. For example, some experimental methods do not need (line 23-25).

  1. You should add “monoclonal antibodies” as novel therapeutic agents in line 43-44.

    3.  As for the abbreviations, CCR2, DMF in line 71, SGK1, FGFR1,             ERK1/2, Akt, JNK, Bax, Bim, Bcl-2, Bcl-xL, Noxa, Puma, mTOR,           STAT3, and p38MAPK in Material and Methods are needed to               specify there. Then, you should remove them in your results              (line 183-185, 226,228-230, 240-243).

  1. You should add the antibody of “NF-κB” in Luminex assay of Materials and Methods.

  1. In Figure 1, I think it is easy to see you replace Figure1A, B and C upside and down.

  1. In Figure 3G, were the survival curves conducted according to Kaplan-Meier method, and was difference compared with the log-rank test or another test? I do not find the method.

  1. In Figure S3B, I think the result of right side bar is not correct. Is a plus and minus each of bortezomib and ixazomib opposite?

  1. The information about supplemental materials are short in the main text, for example there is no material and methods about siRNA on bortezomib or ixazomib resistant, and ARH770 and RPMI8226 are not included in Cell Culture.

Round 2

Reviewer 1 Report

Authors have correctly addressed mot of the issues raised by my previous review. I have no further comments.

Reviewer 2 Report

Dear. Dr. Nishida

 I have reviewed your revised version of manuscript entitled “Activation of serum/glucocorticoid regulated kinase 1/nuclear factor-κB pathway are correlated with low sensitivity to bortezomib and ixazomib in resistant multiple myeloma cells”. The manuscript is well revised, but a few points should be addressed before acceptance for the publication.

Hirono Iriuchishima

  1. You missed upload the supplementary files. So, I could not checked all of your revised manuscript. Please re-upload including them.

  1. You have not answered my first major comment about xenograft model with SGK over-expression. How do you explain about it?

  1. As for the abbreviations, only CCR2 has not been corrected yet (line 79 & 92).

  1. In Figure 2A right, you should mark asterisks. Also, in figure legend of Figure 2B, and 2C, what is the definition of p value?

  1. You had better revise the sentence of line 207-208 as follows. “These analyses revealed that the expression of "several genes" was elevated only in KMS-20 cells. "Above all, we focused on three genes";

  1. In line 327, you should remove Figure 5"B".

Round 3

Reviewer 2 Report

Dear Dr. Nishida.

I have reviewed the latest revised version of manuscript entitled “Activation of serum/glucocorticoid regulated kinase 1/nuclear factor-κB pathway are correlated with low sensitivity to bortezomib and ixazomib in resistant multiple myeloma cells”. The manuscript is well revised, so I think it deserves acceptance for the publication in this journal.